# Dye-Doped Electrically Smart Windows Based on Polymer-Stabilized Liquid Crystal

**DOI:** 10.3390/polym11040694

**Published:** 2019-04-16

**Authors:** Haitao Sun, Zuoping Xie, Chun Ju, Xiaowen Hu, Dong Yuan, Wei Zhao, Lingling Shui, Guofu Zhou

**Affiliations:** 1SCNU-TUE Joint Lab of Device Integrated Responsive Materials (DIRM), National Center for International Research on Green Optoelectronics, South China Normal University, No 378, West Waihuan Road, Guangzhou Higher Education Mega Center, Guangzhou 510006, China; scnush@126.com (H.S.); xiezoping@sina.com (Z.X.); juchun15625136995@163.com (C.J.); yuandong@m.scnu.edu.cn (D.Y.); 20181010@m.scnu.edu.cn (W.Z.); 2Guangdong Provincial Key Laboratory of Optical Information Materials and Technology & Institute of Electronic Paper Displays, South China Academy of Advanced Optoelectronics, South China Normal University, Guangzhou 510006, China; shuill@m.scnu.edu.cn; 3Shenzhen Guohua Optoelectronics Tech. Co. Ltd., Shenzhen 518110, China

**Keywords:** polymer stabilized liquid crystal, smart windows, dye, contrast

## Abstract

Here we report the fabrication of dye-doped polymer-stabilized liquid crystals (PSLC)-based smart windows. The effect of dye doping on PSLC contrast was investigated. Non-dichroic dye tints the PSLC sample in both off- and on-state, which is not beneficial for increasing its off/on contrast. The sample doped with dichroic dye shows a slight color in the off-state and strong color in the on-state, resulting in an enhanced contrast, which attributed to orientation dependent absorption of dichroic dyes. Furthermore, we blended non-dichroic dye and dichroic dye who have complementary absorption together into PSLC mixture. The sample is almost colorless in the off-state due to the subtractive process, while colored in the on-state. The contrast is further enhanced. The results show that the proposed multi-dye-doped PSLC device has high visual contrast and fast response time, making it attractive for applications in light management and architectural aesthetics.

## 1. Introduction

Switchable smart windows, defined as variable light transmittance upon extra potential, are studied worldwide to meet the need for energy saving and other demands in architectures, vehicles, and aircraft [1,2,3,4,5,6,7]. Liquid crystal device is an attractive type of smart windows; in addition to having facile manufacturing process, highly adjustable transmittance, low driven voltage, and fast response speed [1,8,9,10], it is more important to manage the light in the whole visible spectrum. To date, polymer dispersed liquid crystals (PDLC) smart windows are one of the commercialized liquid crystals smart windows [4,11,12,13,14,15]. However, due to the dense polymer network, the anchoring force inhibits the reorientation of the liquid crystals (LC) molecules, which results in a high driving voltage for operating PDLC films. Moreover, the transparency in its clear state still needs further improvement [16]. An alternative to PDLC is polymer-stabilized liquid crystals (PSLC). In this system, the LC materials represent the continuous matrix, while a small amount of polymer network (typically a few percents in weight) is dissolved in the anisotropic fluid [17,18,19]. Due to the smaller polymer concentration, PSLC-based smart windows exhibit ultra-high transparent state and lower driving voltage compared with PDLC film. However, the relatively week haze of PSLC in the scattering state, that is, 40–50%, significantly limits its contrast [20]. C. L. Meng et al. recently reported a type of reverse mode PSLC window by introducing an inhomogeneous alignment surface [21]. The haze of this window in the voltage on-state can be enhanced, while its contrast is still lower than its counterpart PDLC-based windows. 

In this work, we proposed high contrast smart windows combining PSLC with organic dyes. We found that non-dichroic dye tints the PSLC sample in both off- and on-state, which is not beneficial for increasing its off/on contrast. However, the sample doped with dichroic dye shows a slight color in the off-state and strong color in the on-state, resulting in an enhanced contrast, which attributed to orientation dependent absorption of dichroic dyes. Furthermore, we blended two dyes who have complementary absorption together into PSLC mixture. The sample is almost colorless in the off-state due to subtractive process, while colored in the on-state. The contrast is further enhanced. Our results demonstrate that dye doping can enhance the contrast of PSLC smart windows, which also bears out a feasible way to fabricate colored PSLC smart windows by doping mixed dyes for the application of aesthetic aspect.

## 2. Materials and Methods 

### 2.1. Materials

The materials used in this study was commercially obtained, including liquid crystal with negative dielectric anisotropy HNG30400-200 (Δn = 0.149, Δε = −8.3), diacrylate monomer HCM 009, dichroic dye RL002 (purchased from Jiangsu Hecheng Display Technology Co., Ltd., Nanjing, China), anthraquinone dye LT1641B (purchased from Shanghai Lantu Technology Co., Ltd., Shanghai, China), as well as photo-initiator Irg651 (purchased from Tianjin Heowns Biochem LLC, Tianjin, China). The molecular structures of HCM009, dichroic dye, and Irg651 are shown in Figure 1. The structure of anthraquinone dye LT1641B is not provided by its supplier. All materials were not further processed. The composition of the PSLC samples used in this study is shown in Table 1.

### 2.2. PSLC Device Preparation

Firstly, the cleaned indium tin oxide (ITO) coated glasses were irradiated by ozone for 20 min in an ozone tank (BZS250GF-TC, Huiwo) to make it more hydrophilic. Then, Polyimide (PI) (DL-4018, purchased from Shenzhen Dalton Electronic Material Co., Ltd., Shenzhen, China) was spin-coated on ITO glass by a spin coater (KW-4A, Saidekaisi, Setcas electronics Co., Ltd., Beijing, China) at 2500 r for 60 s, and the sample was then heated on a hot plate at 90 °C for 90 s, 105 °C for 30 min, 230 °C for 1.5–2 h under nitrogen atmosphere in sequence. Two obtained ITO glasses with PI layer were placed face to face and was bonded by the mixture of spacers (SiO_2_) and UV glue with a weight ratio of 1:99. The gap between two transparent electrodes was 5 μm decided by spacers. Under UV irradiation for 1 min, a liquid crystal cell was obtained.

The LCs mixture was stirred thoroughly at 60 °C to ensure uniformity before use. The mixture was filled in the cell by capillary force. Then, the cell containing liquid crystals was heated for 30 min at 60 °C followed by natural cooling down to 30 °C. The sample was then irradiated by ultraviolet light with an intensity of 27 mW/cm^2^ for 3 min to induce polymerization. The whole preparation process was conducted in the yellow-light area.

### 2.3. Characterization

Optical transmittance and absorbance were measured using an integrating sphere system (Ocean Optics, Florida, USA). All of the transmittance data presented here were normalized to the substrate. The response time of the samples was measured by a photodiode (FSD1010, Thorlab, Inc., Newtown, NJ, USA) with a time resolution of 65 ns. The output signal of photodiode was amplified and then transformed by an acquisition card (USB6002, National Instruments, Austin, TX, USA), then collected using a LabVIEW program (The project was written by ourself) in the computer.

## 3. Results and Discussion

A cross-section of the PSLC cell is schematically shown in Figure 2a. PSLC layer doped with dyes was sandwiched by two transparent electrodes. The spacer was used to control the cell gap. A vertical alignment layer consisted of polyimide (PI) was coated on the transparent electrodes. The operating principle behind PSLC-based switchable smart windows involves the use of an alternating electric field to actuate PSLC film sandwiched between the two parallel transparent electrodes. Basically, in the voltage-off state, the LC directors exhibit a homeotropic state under the effect of vertical orientation polyimide alignment layer and are stabilized by the formed polymer network through the LC cell (Figure 2b). The incident light is transmitted, and the cell is transparent. In the voltage-on state, the LC directors with negative dielectric anisotropy tend to reorient and deviate from the vertically aligned position. LC molecules in the vicinity of the polymer strands keep their original orientations, while the bulk LC molecules reorient to various azimuthal and polar angles, resulting in randomly aligned poly-domain state (Figure 2b). The incident light is then strongly scattered due to the discontinuity of the refractive index between domains.

To examine the feasibility of the PSLC, a non-dye-doped PSLC smart window was fabricated with the mixture 1. In the off-state, the smart window is clear with the transmittance higher than 95 % in the whole visible range (Figure 2c). A background pattern under the cell is clearly visible, confirming a good transparent state (Figure 2d). On application of the AC voltage, the transmittance drops down. For example, at 30 V, the transmittance decreases to around 10 % in the whole visible range (Figure 2c). The sample switches to a prominent scattering state, and the background pattern can’t be observed anymore (Figure 2d).

To investigate the effect of the polymer concentration on the electro-optical properties of the PSLC cells, several samples containing different concentrations of diacrylate LC monomer were prepared. The transmittance spectra were measured while incrementally increasing the AC voltage. The transmittance-voltage (T-V) curves of the samples are shown in Figure 3a. When the monomer concentration is 1%, the on-state transmittance is high, resulting in a low off/on contrast. When 3% monomer was used, the transmittance of the cell decreases along with the increased voltage. This suggests poly-domain scattering resulting from the reorientation of LC directors under an applied voltage. As the monomer concentration is increased from 3 to 9%, a larger threshold voltage (V_th_) is required to switch on the cell, along with a larger saturation voltage (V_sat_), which is clearly exhibited in Figure 3b. Here, the V_th_ and V_sat_ are, respectively, defined as the voltages at which 90% and 10% of the maximum transmittance of the cell in its voltage ramp-down transmittance curve [22]. The increase of V_th_ and V_sat_ might be due to the denser polymer network in the cell prepared with high monomer concentration, giving rise to an enhanced anchoring force, which affects the reorientation of the LC directors strongly against the applied voltage [23,24].

To enhance the contrast and visibility of smart windows, we doped dyes into PSLC mixture [12]. Figure 4a shows the absorption spectra of selected anthraquinone dye LT1641B. Absorption measurement shows that dye LT1641B has a split absorption peak at 600 and 650 nm exhibiting a blue tint. A few amounts of dye LT1641B was directly dissolved into PSLC mixture (mixture 2) and then filled into the LC cell. Figure 4b shows the transmittance spectra of the colored PSLC cell. In the off-state, the transmittance of the colored PSLC sample has a dramatical reduction at the characteristic absorption peak position of dye LT1641B. The sample is then transparent with vivid color. In the on-state, the transmittance drops down due to scattering. The sample becomes opaque and keeps the color simultaneously (Figure 4c).

Dye LT1641B tints the smart window in both off- and on-state. In order to realize transparency and colorlessness in the off-state and opaqueness and coloredness in the on-state, a dichroic dye RL002 was introduced into PSLC mixture. Figure 5a presents the absorption of dye RL002, which has an absorption peak at 475 nm exhibiting orange tint. The absorption of incident light that is polarized parallel and perpendicular to the principal axes of the dichroic dye is maximal and minimal, respectively. Accordingly, the orientation and absorption of dichroic dye-doped LC cells vary with their alignment, so the absorption is easily modulated by controlling the LC directors. We fabricated PSLC samples with dye RL002-doped PSLC mixture (Mixture 3). In the off-state, the principal axes of the dichroic dye experience the same alignment with LC directors, which is perpendicular to the polarized incident light. This results in weak absorption. Therefore, the transmittance of the sample in the off-state shows a mild drop at the characteristic absorption peak position of dye RL002 (Figure 5b). Correspondingly, the sample exhibits a slight color in the off-state (Figure 5c). In the on-state, the principal axes of dye RL002 reorient along with LC directors under the electric field to be parallel to the polarized incident light, which results in a strong absorption. Therefore, the sample exhibits a strong color in the on-state (Figure 5c). The visual contrast is enhanced.

In order to further eliminate the color of dye RL002-doped PSLC cell in the off-state, we blended dye LT1641B and dye RL002 together into PSLC mixture (Mixture 4), given their complementary absorption. The color of mixed dyes is governed by a subtractive process [25]: any wavelength strongly absorbed by either dye will be absent from the light reflected by the mixture. Therefore, the color is determined by the wavelengths that neither dye absorbs. We prepared a multi-dye-doped PSLC smart window with mixture 4. Figure 6a shows the transmittance of the multi-dye-doped PSLC cell. In the off-state, the transmittance of the sample shows an obvious decrease at the characteristic absorption peak position of dye LT1641B and dye RL002. The tint of dye RL002 is subtracted by the tint of dye LT1641B due to their complementary absorption. The sample is clear and almost colorless (Figure 6b). In the on-state, the absorbance of dye RL002 increases due to the principal axes of dye RL002 realigning to be parallel to the polarized incident light, while the absorbance of dye LT1641B remains unchanged. Therefore, the sample becomes opaque and colored (Figure 6b).

The photo-electrical response of the multi-dye-doped PSLC smart windows to 50 Hz sinusoidal alternating current was also studied. Figure 7a,b are the response time of dye-doped PSLC device with LT1641B:RL002 = 1:2. When the voltage is not applied, the PSLC sample is transparent. After applying a voltage of 30 V, the sample turns into an opaque state after 4 ms (Figure 7a). After removing the applied voltage, the sample turns back to transparent in 24 ms (Figure 7b). 

To explore the switching reliability of PSLC smart windows, we prepared the PSLC sample with mixture 1. The sample was subjected to an applied voltage of 30 V and switched on and off every 3 s by a relay control system. The switch reliability of the sample is illustrated in a plot of transmittance as a function of switching times, as shown in Figure 7c. The unchanged transmittance of the smart window in the off-state and on-state indicates that the window is stable over the course of one hundred thousand times switching.

In order to understand the difference between PSLC smart window and other LC smart windows intuitively, we compared the typical operating parameters of PSLC smart window with the common PDLC smart window, presented in Table 2. On the one hand, the PSLC smart window has a low threshold and saturation voltage. PDLC smart windows could also achieve a low threshold voltage while always keeping a higher saturation voltage. On the other hand, switching-on time for both PSLC and PDLC smart windows is less than 5 ms, which is pretty fast, and switching-off of PSLC smart window is faster than that of PDLC smart windows.

## 4. Conclusions

We fabricated a high-performance multi-dye-doped PSLC smart windows and investigated the effect of dye doping on its optoelectronic properties. Non-dichroic dye tints the PSLC sample in both off- and on-state, which is not beneficial for increasing its off/on contrast. The sample doped with dichroic dye shows a slight color in the off-state and strong color in the on-state, resulting in an enhanced contrast, which attributed to orientation dependent absorption of the dichroic dye. Mixed dyes who have complementary absorption can eliminate the color of PSLC sample in the off-state by the subtractive process, which further enhanced the contrast. Our results demonstrated a feasible way to fabricate colored PSLC smart windows having high visual contrast and fast response time, making it attractive for applications in light management and architectural aesthetics.

## Figures and Tables

**Figure 1 polymers-11-00694-f001:**
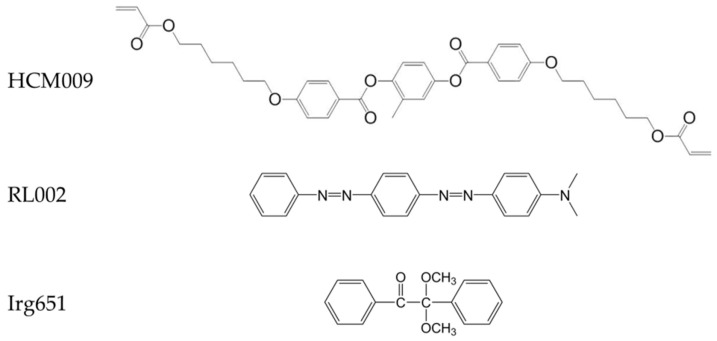
The molecular structure of HCM009, RL002, and Irg651.

**Figure 2 polymers-11-00694-f002:**
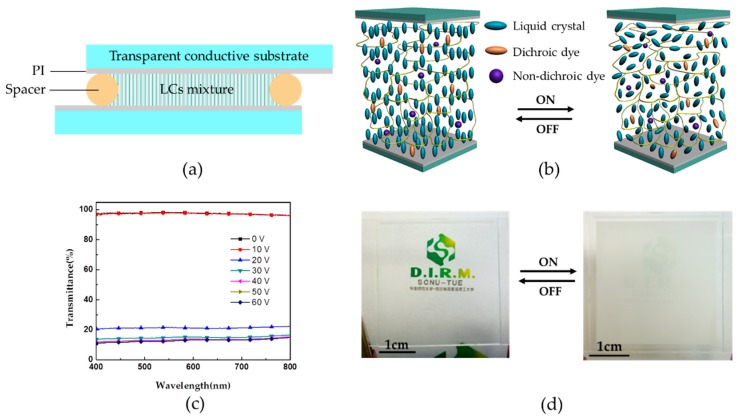
(**a**) Device structure of polymer-stabilized liquid crystals (PSLC) cell (cross-section); (**b**) Schematic illustration of PSLC at voltage-off state and voltage-on state; (**c**) Transmittance of PSLC cell at different applied voltage; (**d**) The photograph of the PSLC cell at the off-state and the on-state (30 V). (Size, 4 × 4 cm). PI: Polyimide; LCs: Liquid crystals.

**Figure 3 polymers-11-00694-f003:**
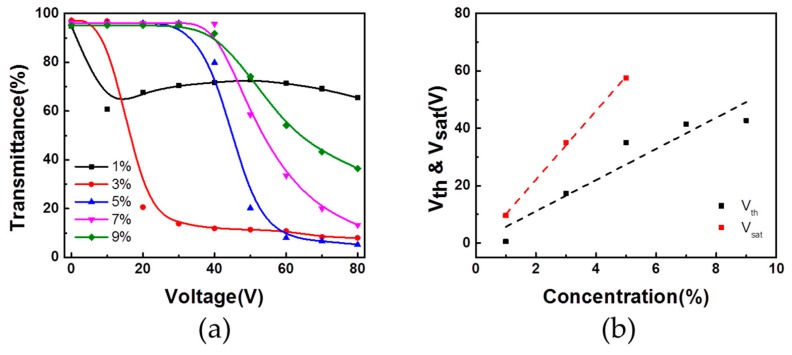
(**a**) Transmittance of polymer-stabilized liquid crystals (PSLC) device based on different monomer concentration; (**b**) Threshold voltage and saturation voltage of PSLC devices based on different monomer concentration. V_th_: larger threshold voltage; V_sat_: larger saturation voltage.

**Figure 4 polymers-11-00694-f004:**
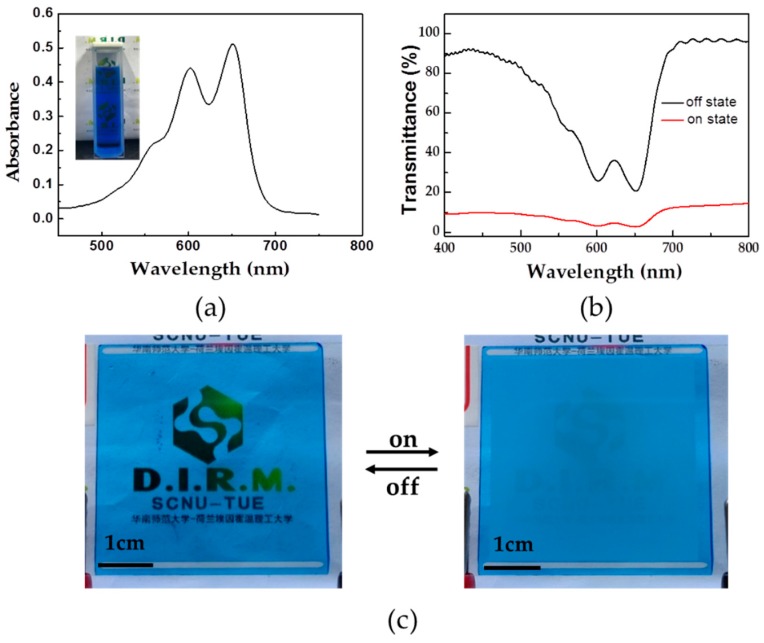
(**a**) Absorption spectra of dye LT1641B. The inset photographs show the color of Dye LT1641B dissolved in dichloromethane; (**b**) Transmittance of dye LT1641B-doped polymer-stabilized liquid crystals (PSLC) sample in the off- and on-state; (**c**) The photograph of the PSLC sample in the off- and on-state (30 V). (Size, 4 × 4 cm).

**Figure 5 polymers-11-00694-f005:**
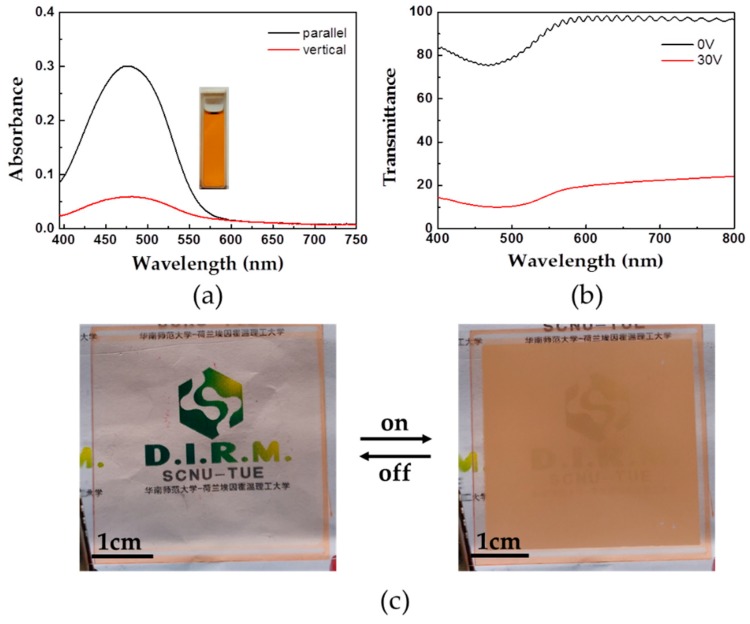
(**a**) Absorption spectra of the dye RL002. Red: the absorption of incident light that is polarized parallel vertical to the principal axes of the dye RL002, Black: the absorption of incident light that is polarized parallel to the principal axes of the dye RL002. The inset photographs show the color of the dye RL002 dissolved in dichloromethane; (**b**) Transmittance of dye RL002-doped polymer-stabilized liquid crystals (PSLC) sample in the off- and on-state; (**c**) The photograph of the PSLC sample in the off- and on-state (30 V). (Size, 4 × 4 cm).

**Figure 6 polymers-11-00694-f006:**
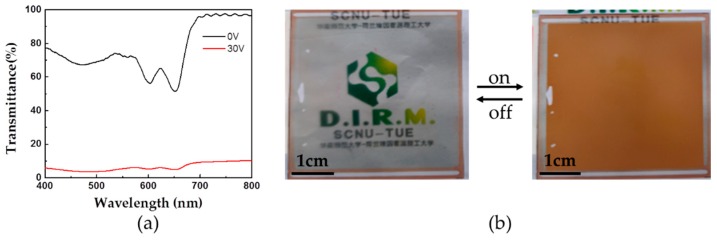
(**a**) Transmittance of mixed dye-doped polymer-stabilized liquid crystals (PSLC) sample in the off- and on-state; (**b**) The photograph of the PSLC sample in the off- and on-state (30 V). (Size, 4 × 4 cm).

**Figure 7 polymers-11-00694-f007:**
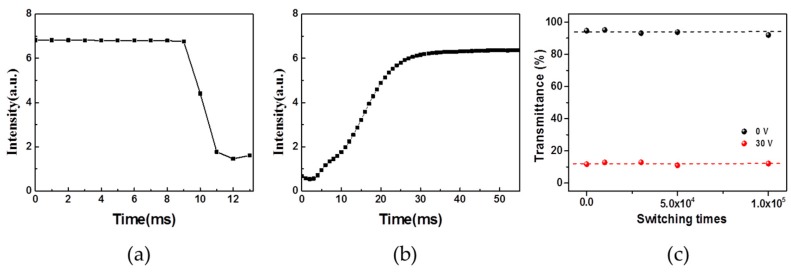
(**a**,**b**) The response time of polymer-stabilized liquid crystals (PSLC) device; (**c**) The transmittance of the PSLC device as a function of cycle times.

**Table 1 polymers-11-00694-t001:** The compositions of the samples studied.

Sample	HCM009	Irg651	LT1641B	RL002	HNG30400-200
	(wt %)	(wt %)	(wt %)	(wt %)	(wt %)
Mixture 1	3	0.5	0	0	96.5
Mixture 2	3	0.5	1	0	95.5
Mixture 3	3	0.5	0	1	95.5
Mixture 4	3	0.5	0.2	0.4	95.9

**Table 2 polymers-11-00694-t002:** Comparison of key parameters of polymer-stabilized liquid crystals (PSLC) smart window and polymer dispersed liquid crystals (PDLC) smart window.

Sample	V_th_ ^1^ (V)	V_sat_ ^2^ (V)	τ_on_ ^3^ (ms)	τ_off_ ^4^ (ms)
PSLC	17.2	35.1	4	24
PDLC [26]	11.1	60	-	-
PDLC [14]	24.6	42.4	1.2	29.3
PDLC [27]	37	80		
PDLC [28]	10	60		193

^1^ V_th_ is threshold voltage; ^2^ V_sat_ is saturation voltage; ^3^ τ_on_ is switching-on time; ^4^ τ_off_ is switching-off time.

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
