# Peer review of "Dye-Doped Electrically Smart Windows Based on Polymer-Stabilized Liquid Crystal"

_polymers, 2019, doi:10.3390/polym11040694_

Round 1

Reviewer 1 Report

Is attached as file

Author Response

We appreciate the reviewer for his/her fruitful comments and thank again for the time and efforts in improving our manuscript. And our responses to the reviewer’s comments from point to point are attached.

Reviewer 2 Report

The paper describes the fabrication and the optical characterization of different polymer stabilized liquid crystals (PSLCs) and their response to an alternating electric field. The author shows that using different dyes as dopant it is possible to switch the material from a transparent to a colored high opaque state opening to their possible application in the smart window field.

The paper is clear and well written. However, before the publication I recommend to modify the manuscript by replying the following questions:

1)      Adding further details on the material preparation. At the moment, reproduction of the experiment is not easy to understand. I suggest to add a Figure with the structure of the molecules as well as their name (or at least commercial acronym). What are HCM009, dye1 and dye 2? What PI has been used for the cell preparation? What lamp (and power) do you use for the polymerization?

2)      To better understand the difference between the material presented and the standard PDLC (used for smart windows), the authors should add some literature data to compare the two systems. E.g. what are the typical voltage and switching time for PDLC?

3)      Since the main difference between PDLC and polymer stabilized liquid crystals is the polymer amount, the author should provide a more detailed study on the effect of the polymer % inside the material (at least for the mixture without dyes). What happen if we use different amount of diacrylate? Is there a critical concentration to decrease/inhibit the material performance?

Sincerely

Author Response

(The authors gave the same response as above.)

Round 2

Reviewer 2 Report

In the revised version, the authors successfully addressed all the points proposed by the reviewers. The new version contains new details on the fabrication as well as on material characterization by changing the polymer concentration.

An interesting comparison with other systems based on LCs for smart windows was added in Table 2.

I recommend the publication of the manuscript without further revision.